# Classification of Tetanus Severity in Intensive-Care Settings for Low-Income Countries Using Wearable Sensing

**DOI:** 10.3390/s22176554

**Published:** 2022-08-30

**Authors:** Ping Lu, Shadi Ghiasi, Jannis Hagenah, Ho Bich Hai, Nguyen Van Hao, Phan Nguyen Quoc Khanh, Le Dinh Van Khoa, Louise Thwaites, David A. Clifton, Tingting Zhu

**Affiliations:** 1Department of Engineering Science, University of Oxford, Oxford OX1 3PJ, UK; 2Oxford University Clinical Research Unit, Ho Chi Minh City 700000, Vietnam; 3Hospital of Tropical Diseases, Ho Chi Minh City 700000, Vietnam; 4Hthe Oxford Suzhou Centre for Advanced Research, University of Oxford, Suzhou Dushu Lake Science and Education Innovation District, Suzhou 215123, China

**Keywords:** tetanus, spectrogram, electrocardiogram, classification, convolutional neural network, channel-wise attention

## Abstract

Infectious diseases remain a common problem in low- and middle-income countries, including in Vietnam. Tetanus is a severe infectious disease characterized by muscle spasms and complicated by autonomic nervous system dysfunction in severe cases. Patients require careful monitoring using electrocardiograms (ECGs) to detect deterioration and the onset of autonomic nervous system dysfunction as early as possible. Machine learning analysis of ECG has been shown of extra value in predicting tetanus severity, however any additional ECG signal analysis places a high demand on time-limited hospital staff and requires specialist equipment. Therefore, we present a novel approach to tetanus monitoring from low-cost wearable sensors combined with a deep-learning-based automatic severity detection. This approach can automatically triage tetanus patients and reduce the burden on hospital staff. In this study, we propose a two-dimensional (2D) convolutional neural network with a channel-wise attention mechanism for the binary classification of ECG signals. According to the Ablett classification of tetanus severity, we define grades 1 and 2 as mild tetanus and grades 3 and 4 as severe tetanus. The one-dimensional ECG time series signals are transformed into 2D spectrograms. The 2D attention-based network is designed to extract the features from the input spectrograms. Experiments demonstrate a promising performance for the proposed method in tetanus classification with an F1 score of 0.79 ± 0.03, precision of 0.78 ± 0.08, recall of 0.82 ± 0.05, specificity of 0.85 ± 0.08, accuracy of 0.84 ± 0.04 and AUC of 0.84 ± 0.03.

## 1. Introduction

Infectious diseases remain a common cause of morbidity and mortality among people living in low- and middle-income countries [1,2,3]. Tetanus disease is an infection caused by a toxin produced by the *Clostridium tetani* bacteria [4]. This powerful neurotoxin inhibits transmission at central synapses resulting in muscle stiffness and spasms and in severe cases, cardiovascular system instability. These symptoms generally progress over a period of 2–5 days. Approximately half of all patients progress to severe disease where spasm control necessitates paralysis and mechanical ventilation. The most common cause of death in settings with access to mechanical ventilation is autonomic nervous system (ANS) dysfunction, occurring in approximately 25% of all patients. Therefore, the early detection of those likely to have severe disease requiring mechanical ventilation or ANS dysfunction is highly valuable as it enables timely intervention and allows appropriate resource allocation [5,6].

The most widely used system for tetanus severity classification is the Ablett score, ranging from 1 to 4. While grades 1 and 2 describe mild or moderate disease progressions where a mechanical ventilation is typically not required, grades 3 and 4 represent severe disease requiring mechanical ventilation. Grade 4 is the most severe form, in which patients have signs of ANS dysfunction [2,7]. Similar to other infectious diseases, early and accurate diagnosis of tetanus severity is extremely important to improve both short- and long-term patient outcomes [8,9,10]. However, in the low- and middle-income countries where tetanus most commonly occurs, facilities and equipment for treatment are often limited. Experienced doctors and nurses have limited time to frequently monitor patients with tetanus. Heart rate variability (HRV) has been shown to be valuable in ANS detection [11,12].

To improve the clinical outcomes and disease incidence of tetanus, we aim to develop a severity warning tool. This tool will use the patient’s electrocardiogram (ECG) data to classify disease severity, aiming to function as a screening tool and guide to clinicians. Such a tool will predict the severity of the disease to help clinicians determine whether the patient must undergo close monitoring and start planning for admission to an intensive care unit (ICU). If the predicted symptoms are mild, the patient can be monitored less frequently in a normal ward. Such a tool would be of particular value for inexperienced or overloaded staff, prevent unnecessary ICU admissions and reduce treatment delays.

In this paper, we design scientific steps as follows: Firstly, the physiological data - electrocardiogram (ECG) data - are collected from the low-cost wearable monitors. Secondly, we propose a warning tool with a deep learning approach for the diagnosis of infectious disease (e.g., tetanus) patients. The aim of this tool is to classify the tetanus severity level, represented as the Ablett score, based on this low-cost ECG data. The contributions of this work are as follows:To the best of our knowledge, this is the first attempt to exploit deep learning with a channel-wise attention mechanism in tetanus diseases detection, which models the channel relationship and boosts the performance of a network. Since the method is completely data-driven, this concept could be transferable to similar infectious diseases.We demonstrate the effectiveness of the proposed method on the low-cost ECG data. We show that our novel method outperforms the sequential techniques. The sequential techniques, including the time-dependent versions of the attention-based network, do not work on low-cost ECG data because the noise of the low quality data disturbs time series analysis.We explore the robustness of the proposed method for the minimal window length of the log-spectrogram.

The paper is structured as follows: Section 2 introduces related work in the diagnosis of tetanus diseases in low- and middle-income countries and deep learning approaches in imaging fields related to our work. Section 3 describes the proposed approach for the tetanus diseases classification. Section 4 provides the details of the tetanus dataset, implementation details, a comparison of baseline methods and the evaluations of the classification results with several performance metrics. Section 5 presents and discusses the experimental results. Finally, Section 6 provides the conclusion of our work.

## 2. Related Work

The early diagnosis of lethal infectious diseases plays an important role in patient treatment. Heart rate is controlled by the ANS and heart rate variability (HRV), i.e., the beat-to-beat changes in RR intervals, is linked to changes in ANS activity [11,12]. In tetanus, disease severity is associated with ANS activity, and changes in conventional HRV parameters measured from ECG have been shown to correlate with disease severity. To evaluate autonomic nervous system disturbance (ANSD), the HRV-based methods need an extra preprocessing step and then require features such as RR intervals and QRS complex extraction [6,13,14]. Lin et al. [15] use HRV as an index to detect the disease progression which is caused by enterovirus infection. However, it is still a challenge to robustly extract RR intervals.

Recently, machine learning approaches have been used to diagnose and classify severity of infectious diseases. Traditional machine learning approaches require the feature engineering process for manually selecting and transforming features from the dataset. Tadesse et al. [16] applied support vector machines (SVM) to automatically detect the ANS dysfunction level for hand, foot and mouth disease (HFMD) and tetanus. They also demonstrated SVM outperforms HRV on infectious diseases detection. Tadesse et al. [17] use spectrograms of ECG and PPG with transfer learning to classify severity of two infectious diseases, tetanus and HFMD, and prove deep learning methods outperform traditional machine learning methods (e.g., SVM). Kiyasseh et al. [18] suggest generating pathological photoplethysmogram (PPG) signals to boost diagnosis performance (e.g., tetanus and HFMD). The previous works from [16,17,18] study a small dataset of tetanus; consequently, their results are limited.

One prominent advantage of convolutional neural networks (CNNs) is their capability to implicitly learn to extract relevant features. The one-dimensional (1D) CNNs have been widely used in signal processing applications, such as biomedical data classification and early diagnosis [19,20]. In order to use the 2D CNN, the common approach transforms the 1D signal to a 2D representation by time–frequency analysis, such as spectrogram, log spectrogram, mel spectrogram and scalogram [17,21,22,23,24], and the 2D representation obtained is considered an image. The spectrogram provides a visual presentation of dynamic information which can be composed of low-level features such as lines and edges [17]. Based on the recent literature work, we know the 2D CNN performs well in image classification. Using 2D spectrograms, an image-based ECG signal classification structure achieves a better performance than the 1D CNN [25].

Computer vision and image analysis have been revolutionized by the attention mechanism [26]. The benefits of the attention mechanism range are across different topics, from image classification [27,28] to action recognition [29,30], for improving representation power of networks. There are two families of attention mechanisms in deep learning, soft attention and hard attention. In soft attention, the features of the image are multiplied with a mask of values between zero and one. In hard attention, the deep learning model focuses on the input information from a small portion of the whole image, with a mask of values of zero or one. In general, attention is implemented by a combination with a gating function, such as a sigmoid or softmax. Attention can also be implemented via combinating with sequential techniques, such as long short-term memory (LSTM) [31]. So far, the attention mechanism has not been implemented in infectious diseases for improving diagnostic accuracy. Therefore, it is a novel application of the attention mechanism to an ECG dataset acquired from patients with infectious diseases. In previous studies, spectrograms of tetanus with transfer learning [17] do not consider which part of the feature maps is more important. Our work will use soft self-attention to weight the channel-wise responses in the convolutional layers for modelling inter-dependencies among the channel-wise features.

## 3. Method

The proposed framework includes the following steps:*Data Preprocessing*: ECG noise removal;*Spectrogram analysis of single-lead ECG signal*: Generated 2D log-spectrograms as inputs of the proposed method;*Feature extraction with CNN*: Feed the log-spectrograms into convolutional layers to extract features;*Attention Mechanism*: Model the inter-dependencies among the channel features of the convolutional layers.

The proposed method is named 2D-CNN + Channel-wise Attention. To understand how the proposed method makes a decision and what the network sees in each layer in the method, we explore the visual explanation algorithm-gradient-weighted class activation mapping (Grad-CAM) [32]-in the proposed method. Figure 1 shows the overview of the proposed method and the visual explanations of features in the last layers of 6 convolutional blocks of the method.

### 3.1. Data Preprocessing

There are mainly two types of noise that influence the ECG signal. Firstly, patient muscle movement causes low band frequency noises [33]. Secondly, the electrical source which operates the ECG monitor leads to high band frequency noise [33]. Given low-cost ECG signal, we use a one-lead ECG from an ECG signal and perform preprocessing to clean the data and remove the background noise from the input ECG signal using a Butterworth filter. We set a cutoff point of 0.05 Hz for the high-pass filter and 100 Hz for the low-pass filter. The implementation is performed utilizing the SciPy package [34].

### 3.2. Logarithmic Spectrogram Generation

A 1D ECG is not able to use 2D CNNs. If the ECG is represented in 2D, such as an image, we can use the successful approaches in image classification to deal with the signal. Hence, we transform the preprocessed ECG into spectrograms. The spectrogram is a 2D time–frequency representation based on the consecutive Fourier transform. The logarithmic spectrogram is a log-scaled spectrogram based on the consecutive Fourier transform, and it pays more attention to lower frequencies. Next, we normalise the spectrograms by their maximum value and scale the value in the range 0 to 255 and logarithmic scale of the normalised spectrograms (see Equation (Equation 1)).
(1)$$\tilde{V}=\log\left(\frac{V}{max(V)}*255\right)$$

Figure 2 shows examples of spectrograms and normalised logarithmic spectrograms of ECG with mild and severe symptoms. The visible image patterns in spectrograms are hard to see. By using a log scale, the low frequencies are easier to see as log-spectrograms. Hence, the use of log-spectrogram image patterns enhances understanding of 2D CNNs.

### 3.3. Attention-Based Network

#### 3.3.1. Convolutional Layers

The convolutional layers explore the spatial information in each 2D spectrogram (intra-slice information). The architecture of each block was inspired by Zihlmann et al. [22] and consists of the convolutional blocks of the 2D convolutional layers (3×3 kernel size), ELU and 2D batch normalization. The second, fourth and sixth convolutional blocks are followed by a 2D max pooling layer (2×2 window), respectively (See Figure 1). A logarithmic spectrogram is input to the convolutional layers.

#### 3.3.2. Channel-Wise Attention

Changing the weight of the different channels in the feature maps, the proposed model can emphasise the most important features and suppress less useful features. Hence, the channel-wise attention mechanism can explore the relationships of features among different channels and add weights as soft attention for each channel. Inspired by the squeeze-and-excitation networks [28], we add the channel-wise attention at the end of each convolutional block. The illustration in Figure 3 shows the structure of the channel-wise attention mechanism.

Let U = [$u_{1}$, $u_{2}$, …, $u_{n}$], where $u_{i}$∈$R^{W\times H}$ denotes the feature map on the i-th channel, *n* is the number of channels, and the *W* and *H* are width and height of $u_{i}$. For the squeeze operation, we aim to squeeze global spatial information into a channel feature. We apply a 2D adaptive average pooling to obtain a single value for each channel feature (see Equation (Equation 2)). The channel features *m* can be represented as $M=[m_{1},m_{2},\ldots ,m_{n}]$, where $m_{i}\in R^{n}$ denotes the i-th channel features, and $m_{i}$ is the average of the vector $u_{i}$. The channel features can be calculated by
(2)$$m_{i}=\frac{1}{W\times H}\sum_{p=1}^{W}\sum_{q=1}^{H}u_{i}\left(p,q\right).$$

Next, we perform an excitation operation on the single value for obtaining the channel weights that represent channel-wise dependencies. The excitation operation uses a gating function with a sigmoid activation, which can be represented as
(3)$$s_{i}=\sigma \left(W_{2}\phi \left(W_{1}m_{i}\right)\right),$$
where σ and ϕ refer to the sigmoid and ReLU function, respectively. W1 and W2 represent the learnable parameter matrices.

The output of the excitation operation - channel weights - are element-wise multiplied on the output features of each convolutional block. The final output of the channel-wise attention block can be represented as
(4)$$\tilde{x_{i}}=u_{i}⨂m_{i},$$
where ⨂ is the channel-wise multiplication between the feature map $u_{i}$ and the weight vector $m_{i}$.

#### 3.3.3. Loss Function

The binary cross-entropy (BCE) loss function [35] is used in the proposed method, which is defined as
(5)$$L_{BCE}=-\frac{1}{N}\sum_{i=1}^{N}(y_{i}\cdot log\hat{y_{i}}+\left(1-y_{i}\right)\cdot log\left(1-\hat{y_{i}}\right),$$
where $y_{i}$ is the *i*th target label, $\hat{y}$ is the prediction of the *i*th label, and N is the batch size. We combine a sigmoid layer and a BCELoss in one single class. This combined loss function is more numerically stable than using a plain sigmoid followed by a BCELoss; by combining the operations into one layer, we take advantage of the log-sum-exp trick for numerical stability.

After the attention layer of the last convolutional block, we choose 3 fully connected layers that output our 2 labels. The output of the fully connected layer is fed as inputs to the sigmoid layer, and the output of the sigmoid layer is turned into the probability of the tetanus mild and severe classes.

## 4. Experiments

### 4.1. ECG Acquisition for Tetanus Patients

To acquire ECG data from tetanus patients, we use the low-cost wearable monitor ePatch [36] (see Figure 4). The ePatch (ePatch. https://www.myheartmonitor.com/device/epatch/ (accessed on 21 August 2022)) sensor includes all the electronic components: a rechargeable battery, a signal processing component, a data storage component and wireless data transmission equipment [37]. The doctor attaches the lightweight cardiac monitor firmly to the patient’s chest. The ePatch will automatically record the ECG once the system is installed. Figure 2 shows examples of ECG waveforms collected from tetanus patients.

The study data collection has been approved by the relevant ethical committees. This dataset has been published previously [38] and is collected from 110 patients at the Hospital for Tropical Diseases, Ho Chi Minh City, Vietnam. The ECG waveforms from the tetanus patients are collected with a sampling rate of 256 Hz. The first 24 h ECG data is recorded on day 1 when a patient is admitted to the infectious disease department. The second 24 h ECG data is recorded on day 5 of hospitalization.

The dataset used in this study consists of 178 ECG example files from 110 patients on days 1 and 5. We use trimmed ECG data [38] which removes most of the noise from the ECG recording. Then we split our data into the training/validation/test datasets with a 141/19/18 ratio.

The ePatch can collect two-lead ECG signals. We extract the one-lead ECG signal for our experiments. We perform spectrogram analysis of this single-lead ECG signal. We transform the ECG signal to a 2D image, extract features from a 2D CNN and then model inter-dependencies among the channel-wise features. Our deep learning model will classify the ECG signal into two categories; label 0 represents mild tetanus (Ablett grade 1 and 2), and label 1 represents severe tetanus (Ablett grade 3 and 4).

### 4.2. Implementation Details

#### 4.2.1. Data Preprocessing

The time series ECG waveform is divided into a sequence of ECG samples without overlapped windows. We set the duration of the window length as 60 s. We choose 30 60-s ECG samples from each ECG example file. There are 4230 (141 × 30) ECG log-spectrograms in the training set, including 2370 samples of the mild disease and 1860 samples of the severe disease; 540 (19 × 30) ECG log-spectrograms in the validation set, including 270 samples of the mild disease and 270 samples of the severe disease; 570 (18 × 30) ECG log-spectrograms in the test set, including 360 samples of the mild disease and 210 samples of the severe disease.

Spectrograms are computed by scipy.signal.spectrogram in SciPy [34]. We choose the Tukey window width to be 25% of a window’s length overlap. We set the nperseg-length of each segment as 64, and the noverlap numbers of points to overlap between segments as 32. There are 15,360 = 256 Hz × 60 s sampling points in a window of length which are used to compute a spectrogram; these are based on 60 s at the sampling rate of 256 Hz of the ECG data. We then perform normalization and logarithmic scale on the spectrogram (see Figure 2). The spectrogram is saved as a PNG format image with the default ’viridis’ colormap. Finally, the rectangular picture of the spectrogram (479×33 pixels of the log-spectrograms on every 60 s ECG) is ready for the proposed deep learning approach.

#### 4.2.2. Training

The model is trained over 100 epochs using the Adam optimizer with a learning rate 0.001 and a batch size of 32. The mean squared error (MSE) is chosen as the evaluation metric. We choose torch.nn.BCEWithLogitsLoss for the loss function. The proposed network was implemented using Python 3.7 with Pytorch. Experiments are run with computational hardware NVidia GeForce GTX 1080 Ti GPU 10 GB and NVidia GeForce RTX 3060 12 GB.

### 4.3. Baseline Methods

Because the attention mechanism can be implemented by combining it with either the gating function or sequential techniques, we aim to compare these two attention combination style networks. In our work, we compare the proposed 2D-CNN + Channel-wise Attention method with six different methods. Figure 5 shows six 2D deep learning methods: 2D-CNN, 2D-CNN + Channel-wise Attention + ConvLSTM, 2D-CNN + Channel-wise Attention + LSTM, 2D-CNN + LSTM, 2D-CNN + ConvLSTM and 2D-CNN + Dual Attention. Here Channel-wise Attention and Dual Attention belong to the gating function; ConvLSTM and LSTM belong to sequential techniques.

To investigate how the attention mechanism works in the proposed method, we compare the methods with and without attention layers. We also compare the 2D-CNN with the 1D-CNN for testing the image-based ECG signal classification method.

### 4.4. Evaluation Metrics

In the binary classification, the terms true positive (TP), true negative (TN), false positive (FP) and false negative (FN) are used to calculate accuracy, precision, specificity, recall and F1-score [17].
$$Accuracy=\frac{TP+TN}{TP+TN+FP+FN}$$
$$Precision=\frac{TP}{TP+FP}$$
$$Specificity=\frac{TN}{TN+FP}$$
$$Recall=\frac{TP}{TP+FN}$$
$$F1=2*\frac{precision*recall}{precision+recall}$$

We run each model five times and calculate the mean and the standard deviation of the performance metrics on the test dataset.

## 5. Results and Discussion

In this section, we evaluate the proposed method and show how it works. Then, we compare it to the vanilla 2D-CNN as a benchmark. We also present results on the longitudinal data, which are all essentially time-dependent versions of the previously used ones. Moreover, we analyse the method’s parameters regarding more efficient computation. In addition, we compare the proposed method to the traditional machine learning method of random forest. In our experiments, we run each model five times with the same split training/validation/test datasets. We perform the splitting of the dataset into training, validation and test based on unique ECG samples. After splitting, we apply windowing on ECG time series to split each signal into 60 s time series. Therefore, we made sure that ECG samples in each split of the dataset are unique.

### 5.1. Attention Layers

We have investigated different attention mechanisms including spatial attention and channel-wise attention. Due to GPU memory capacity limitations, the self-attention model cannot be tested in our experiments. However, we are able to compare the dual attention model (position attention and channel attention modules) [39] to the proposed 2D-CNN + channel-wise attention model. According to the experimental results in Table 1, we found that the channel-wise attention outperforms the dual attention mechanism. Figure 6 shows the examples of Grad-CAM visual explanations of the features for label 0 - mild tetanus - in all different layers of the baseline 2D-CNN method and the proposed 2D-CNN + Channel-wise Attention method. The different important locations of features are visualised by colours. The red colours emphasise the most important parts where the model focuses on different layers for classification. Compared to the last layer in block six, there are more red areas in the proposed method, meaning that this area influences the final decision for label 0 - mild tetanus.

### 5.2. Sequential Techniques

The different longitudinal models are all essentially time-dependent versions of the previously used ones: 2D-CNN and 2D-CNN + Channel-wise Attention.

#### 5.2.1. Recurrent Neural Network Layers

As shown in Table 1, the 2D-CNN + Channel-wise Attention performs better than the 2D-CNN + Channel-wise Attention + LSTM, and the 2D-CNN performs slightly better than the 2D-CNN + LSTM. The low-cost ECG signal quality is too low to perform longitudinal data analysis with recurrent neural networks. There is a great deal of background noise in the ECG data. Although we performed preprocessing to filter out the noise, the cleaned ECG data still contains noise, which influences the results of the recurrent neural network.

#### 5.2.2. Convolutional LSTMs Model

Inspired by Lu et al. [40], we use a single layer convolutional LSTM (ConvLSTM) [41] to explore the temporal relationships among the three log-spectrograms. As shown in Figure 5, the 2D-CNN + ConvLSTM and 2D-CNN + Channel-wise Attention + ConvLSTM methods are explored in our experiment. The output of the convolutional layers will be the input of the ConvLSTM layer. We set T=3, 60 s for a 20-s window duration. The ConvLSTM makes decisions on the features of three log-spectrogram samples. Table 1 shows that the ConvLSTM models do not perform well in 1-lead ECG data, suggesting that the ECG signal quality in resource-limited settings is too low for temporal information analysis.

### 5.3. 1D Convolutional Model

The 1D-CNN has the same architecture as the 2D-CNN (shown in Figure 5). However, the 1D-CNN model uses 1D convolution instead of 2D convolution at its convolutional layers. As shown in Table 1, the performance of the 1D-CNN is slightly lower than the 2D-CNN using a 60-s window length log-spectrogram without downsampling. Compared to the 2D-CNN + Channel-wise Attention, the 1D-CNN has lower performance metrics. The results show that the image-base method works better, and the channel-wise information boosts the performance of diagnosing tetanus.

### 5.4. Downsample Spectrogram

Due to the computational limits of the GPUs, we aim to develop a deep learning pipeline with low computational cost. Therefore, we perform experiments on downsampled spectrograms using scipy.signal.decimate. We downsample spectrograms four times in the horizontal axis and the vertical axis, respectively. As shown in Table 2, spectrograms without downsampling produce better F1 scores, specificities and accuracies than those with downsampling for 60 s window length spectrograms. Because the downsampled spectrograms are too small, the convolutional LSTM methods - 2D-CNN + ConvLSTM and 2D-CNN + Channel-wise Attention + ConvLSTM - fail. Hence, we suggest using spectrograms without downsampling as inputs in the proposed model.

### 5.5. Misclassification

In the training phase, we run each method five times, which gives five trained models in each method. Next, we obtain five different confusion matrices using the test dataset. Here we average confusion matrix numbers over the five different runs. The confusion matrices in Figure 7 show a holistic view of how well each method in our experiments performs and what kind of misclassification they make between the mild and severe levels. As shown in Figure 7a, the true successful detection of the severe tetanus diagnosis increases from 122 to 171 after employing channel-wise attention layers. It also increases to 129 after employing dual attention layers. Figure 7c shows the 1D-CNN better predicts severe tetanus than mild tetanus, with 162 correct severe tetanus diagnoses out of 210 samples and 253 correct severe mild diagnoses out of 360 samples. Compared to the same method from Figure 7a,b, the correct mild and severe tetanus diagnosis numbers are higher in (a) than (b). This demonstrates the 60 s window length log-spectrograms without downsampling as inputs work better than the downsampled log-spectrograms.

### 5.6. Window Length of Spectrogram

In order to evaluate the robustness of the proposed method (spectrograms without downsampling), we perform experiments on the window duration of the ECG and check the minimal window length of the spectrogram that can still diagnose tetanus. We have investigated the 60 s, 50 s, 40 s, 30 s, 20 s, 10 s and 5 s window lengths of raw ECG data for spectrogram generation. For the experiments, the size of the training/validation/test dataset does not change. As shown in Table 3, the 10 s and 5 s window lengths of the raw ECG data are too short to generate a useful spectrogram for deep learning approaches. Comparisons with the 60 s window length show that the 50 s, 40 s, 30 s and 20 s window length spectrograms can still maintain an accurate tetanus diagnosis.

Figure 8 shows the examples of Grad-CAM visual explanations of the features for mild tetanus in the different layers in the proposed method. The log-spectrograms without downsampling are the inputs of the proposed method. From the visual explanations of features, we can see that the channel-wise attention layer emphasises some parts of the feature image compared to the previous layer in the convolutional block, particularly in the green rectangle area of Figure 8.

### 5.7. Traditional Machine Learning

We compare the proposed 2D-CNN + Channel-wise Attention with the traditional machine learning method random forest [42,43]. The details of extracted features are shown in Table 4, including eight HRV time domain features. There are several open-source toolboxes that compute HRV based on raw ECG signal [44,45,46,47]. In our work, we detect r peaks of ECG using the open-source packages py-ecg-detectors 1.3.2 [48] and extract features using the open-source packages hrv-analysis 1.0.4 [47].

The comparisons are shown in Table 5. The F1 score is higher for random forest with HRV time domain features compared to the proposed 2D-CNN + Channel-wise Attention. The F1 score is the harmonic mean of precision and recall. Precision evaluates how precisely a method predicts severe tetanus (TP). Recall measures the percentage of the correctly predicted severe tetanus (TP) that a method detects. Random forest with HRV time domain features yields a better prediction of severe tetanus.

## 6. Conclusions

We proposed a deep learning method, 2D-CNN with a Channel-wise Attention mechanism, to classify the severity of tetanus using wearable monitors in a resource-limited setting. We cleaned the background noise, and we were able to classify tetanus symptoms as mild or severe tetanus. Despite this, there are limitations to this method. Firstly, the ECG data from the wearable monitors have a much lower signal-to-noise ratio. This makes reducing the large amount of noise from the wearable monitors ECG data a significant challenge. However, the low-cost ECG data are affordable in low- and middle-income countries, and we are able to reliably use this low-quality data. Secondly, due to the small dataset used, to make a classification of tetanus severity requires ECG data recorded on day 1 and day 5. In the future, we will extend the dataset in order to predict the severity of tetanus on day 5 using the ECG data from only day 1.

We investigate the window length of the spectrogram and investigate the range of window lengths that can maintain an accurate tetanus diagnosis. In our experiments, a 50 s window has a relatively higher value of performance metrics than other window lengths. We will explore time series imaging further in future work, which will aim to find the optimal range of time windows.

In future work, we will consider sequence learning via transformers. A combination of CNN and transformer networks are used to extract both local features and global dependencies [49,50,51]. Moreover, we will also apply knowledge distillation techniques to this combination network for further improving the accuracy of tetanus diagnoses. The rationale for using the knowledge distillation is the low processing and sensor costs. Knowledge distillation extracts the knowledge from the large complex teacher model and passes it on to the small simple student model [52,53]. This distilled procedure will not require the training of a large number of tetanus data.

## Figures and Tables

**Figure 1 sensors-22-06554-f001:**
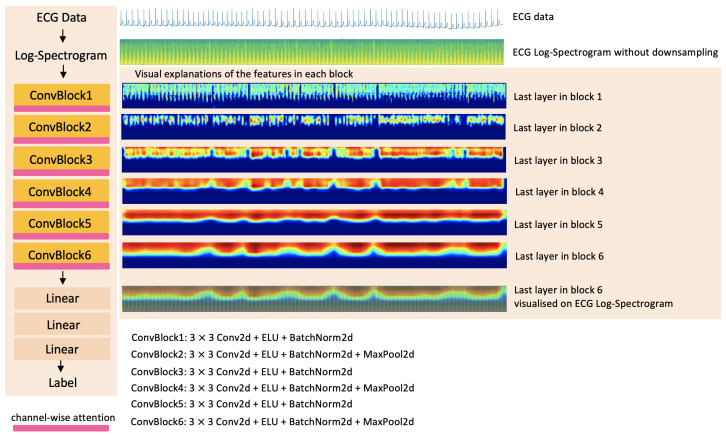
Overview of the proposed method and the visual explanations of features for the label 0-mild tetanus in the last layers of 6 convolutional blocks of the method. The 60-s window length log-spectrogram of raw ECG data is the input of the proposed method called 2D-CNN + Channel-wise Attention. The output of the proposed method is the label prediction, label 0 (mild tetanus) and label 1 (severe tetanus).

**Figure 2 sensors-22-06554-f002:**
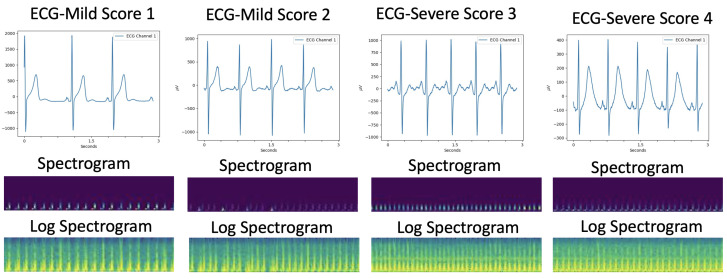
Examples of ECG waveforms collected from tetanus patients, spectrograms and normalised log-spectrograms generated for each tetanus classification: scores 1 and 2 refer to mild symptoms and scores 3 and 4 refer to severe symptoms.

**Figure 3 sensors-22-06554-f003:**
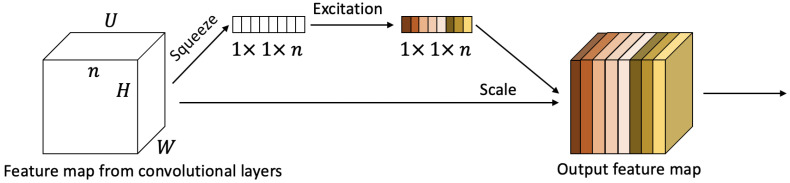
Illustration of the channel-wise attention mechanism structure. The channel-wise attention is added at the end of each convolutional block, which models the interdependencies among the channel features of the convolutional layers.

**Figure 4 sensors-22-06554-f004:**
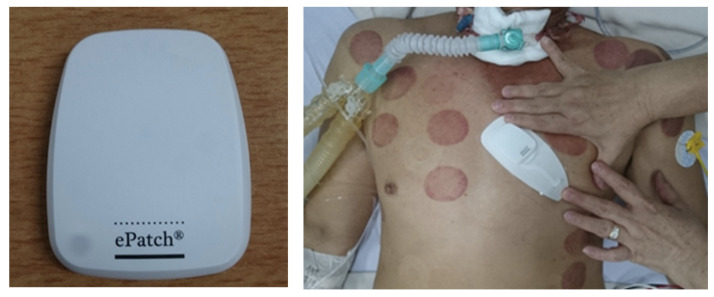
Wearable monitor for ECG data acquisition; ePatch (**left**) and example of ePatch placement on the chest (**right**).

**Figure 5 sensors-22-06554-f005:**
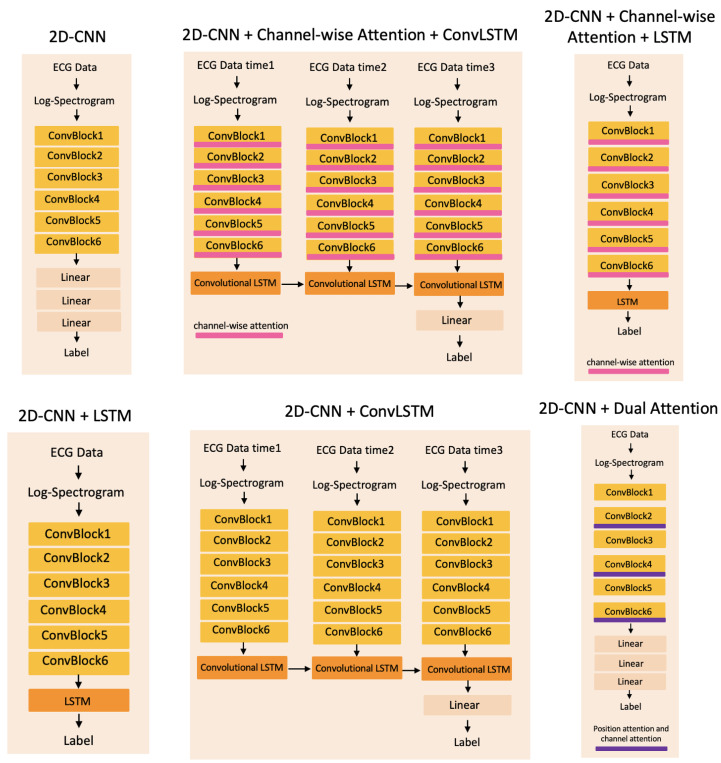
The architecture of deep learning methods which are used to compare with the proposed method. The methods from left to right, top: 2D-CNN; 2D-CNN + Channel-wise Attention + ConvLSTM; 2D-CNN + Channel-wise Attention + LSTM; bottom: 2D-CNN + LSTM; 2D-CNN + ConvLSTM; 2D-CNN + Dual Attention.

**Figure 6 sensors-22-06554-f006:**
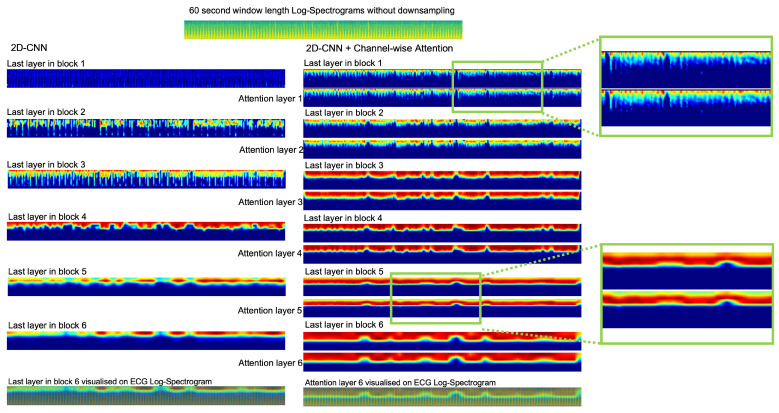
Examples of visual explanations of the features in all different layers of the baseline 2D-CNN method and the proposed 2D-CNN + Channel-wise Attention method. The 60-s window length log-spectrogram of raw ECG data is the input of these two methods. The green rectangle highlights the huge visual difference between the adjacent layers in the proposed method.

**Figure 7 sensors-22-06554-f007:**
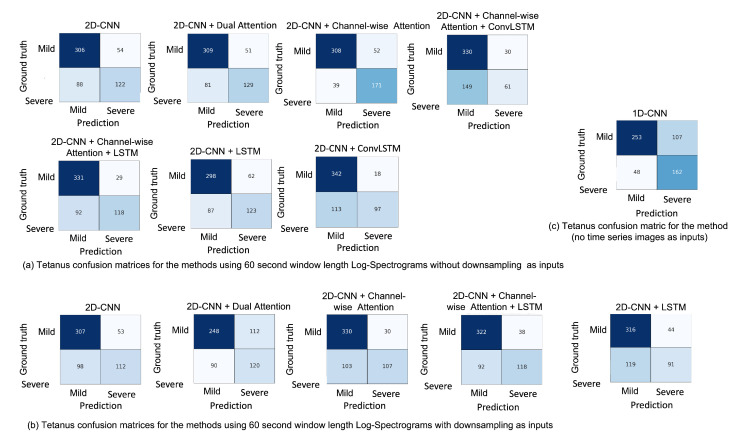
Confusion matrices of the tetanus severity level diagnosis using different deep learning methods: (**a**) tetanus confusion matrices for the methods using 60 s window length log-spectrograms without downsampling as inputs; (**b**) tetanus confusion matrices for the methods using 60 s window length log-spectrograms with downsampling as inputs; (**c**) tetanus confusion matrices for the 1D-CNN method (no time series images as inputs).

**Figure 8 sensors-22-06554-f008:**
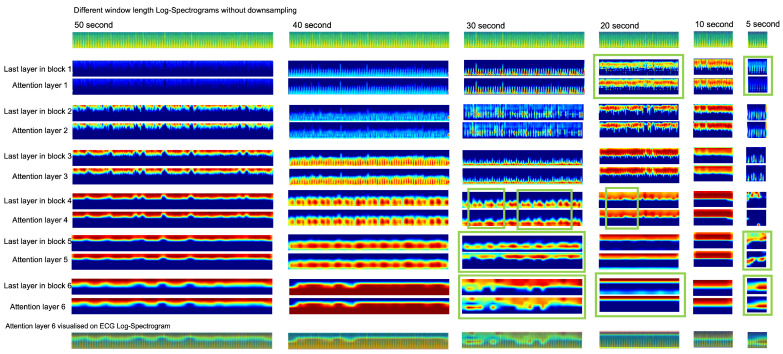
Examples of visual explanations of the features in the last layer of each convolutional block and each attention layer of the proposed method. The log-spectrograms without downsampling are the inputs of the proposed method. These log-spectrograms are generated from the 50 s, 40 s, 30 s, 20 s, 10 s and 5 s window lengths of raw ECG data. The green rectangle highlights the huge visual difference between the adjacent layers.

**Table 1 sensors-22-06554-t001:** Quantitative comparison on the proposed 2D-CNN + Channel-wise Attention method and the baseline methods. The results are presented as mean ± standard deviation. The best performance is indicated in bold.

	**60 s Window Length Log-Spectrogram without Downsampling**					
**Method**	**F1 Score**	**Precision**	**Recall**	**Specificity**	**Accuracy**	**AUC**
2D-CNN	0.61 ± 0.14	0.68 ± 0.07	0.57 ± 0.19	0.85 ± 0.02	0.75 ± 0.07	0.72 ± 0.09
2D-CNN + Dual Attention	0.65 ± 0.19	0.71 ± 0.17	0.61 ± 0.21	0.86 ± 0.09	0.76 ± 0.11	0.74 ± 0.13
2D-CNN + Channel-wise Attention	**0.79 ± 0.03**	**0.78 ± 0.08**	**0.82 ± 0.05**	0.85 ± 0.08	**0.84 ± 0.04**	**0.84 ± 0.03**
2D-CNN + LSTM	0.61 ± 0.15	0.71 ± 0.16	0.59 ± 0.20	0.83 ± 0.17	0.74 ± 0.10	0.71 ± 0.10
2D-CNN + ConvLSTM	0.52 ± 0.32	0.77 ± 0.23	0.46 ± 0.33	0.95 ± 0.04	0.77 ± 0.11	0.71 ± 0.15
2D-CNN + Channel-wise Attention + ConvLSTM	0.38 ± 0.17	0.67 ± 0.10	0.29 ± 0.16	**0.92 ± 0.06**	0.68 ± 0.05	0.60 ± 0.06
2D-CNN + Channel-wise Attention + LSTM	0.59 ± 0.32	0.70 ± 0.34	0.56 ± 0.34	0.92 ± 0.92	0.79 ± 0.12	0.74 ± 0.16
	**No Time Series Images**					
**Method**	**F1 Score**	**Precision**	**Recall**	**Specificity**	**Accuracy**	**AUC**
1D-CNN	0.65 ± 0.14	0.61 ± 0.05	0.77 ± 0.25	0.70 ± 0.13	0.73 ± 0.05	0.74 ± 0.08

**Table 2 sensors-22-06554-t002:** Quantitative comparison on the proposed 2D-CNN + Channel-wise Attention method and the baseline method using downsampled log-spectrogram. The results are presented as mean ± standard deviation. The best performance is indicated in bold.

	60 s Window Length Log-Spectrogram with Downsampling					
Method	F1 Score	Precision	Recall	Specificity	Accuracy	AUC
2D-CNN	0.58 ± 0.16	0.68 ± 0.05	0.53 ± 0.19	0.85 ± 0.06	0.74 ± 0.05	0.69 ± 0.07
2D-CNN + Dual Attention	0.54 ± 0.08	0.57 ± 0.17	**0.57 ± 0.21**	0.69 ± 0.23	0.65 ± 0.09	0.63 ± 0.06
2D-CNN + Channel-wise Attention	0.60 ± 0.10	**0.82 ± 0.10**	0.51 ± 0.16	**0.92 ± 0.08**	0.77 ± 0.30	0.71 ± 0.05
2D-CNN + LSTM	0.52 ± 0.12	0.67 ± 0.03	0.43 ± 0.14	0.88 ± 0.03	0.71 ± 0.04	0.66 ± 0.06
2D-CNN + Channel-wise Attention + LSTM	**0.63 ± 0.13**	0.75 ± 0.05	0.56 ± 0.19	0.89 ± 0.04	**0.77 ± 0.05**	**0.73 ± 0.08**

**Table 3 sensors-22-06554-t003:** Quantitative comparison on window length of log-spectrograms as the inputs of the proposed 2D-CNN + Channel-wise Attention method. The results are presented as mean ± standard deviation. The best performance is indicated in bold.

	The Proposed Method (Spectrograms without Downsampling)					
Window Duration	F1 Score	Precision	Recall	Specificity	Accuracy	AUC
50 s	**0.81 ± 0.05**	0.81 ± 0.06	**0.82 ± 0.04**	0.88 ± 0.04	**0.86 ± 0.04**	**0.85 ± 0.04**
40 s	0.80 ± 0.04	0.84 ± 0.08	0.77 ± 0.07	0.91 ± 0.05	0.86 ± 0.03	0.84 ± 0.03
30 s	0.74 ± 0.05	0.79 ± 0.07	0.79 ± 0.07	0.87 ± 0.06	0.84 ± 0.04	0.83 ± 0.04
20 s	0.79 ± 0.05	0.80 ± 0.08	0.78 ± 0.07	0.88 ± 0.07	0.84 ± 0.04	0.83 ± 0.04
10 s	0.55 ± 0.33	0.74 ± 0.16	0.45 ± 0.38	0.90 ± 0.06	0.77 ± 0.12	0.72 ± 0.17
5 s	0.43 ± 0.32	**0.98 ± 0.02**	0.34 ± 0.29	**0.99 ± 0.01**	0.75 ± 0.10	0.67 ± 0.14

**Table 4 sensors-22-06554-t004:** List of extracted heart rate variability (HRV) features in traditional machine learning.

Parameters	
	HRV time domain features
mean_nni	mean of RR-intervals
sdnn	standard deviation of RR-intervals
sdsd	standard deviation of differences between adjacent RR-intervals
rmssd	square root of the mean of the sum of the squares of differences between adjacent NN-intervals
mean_hr	mean Heart Rate
max_hr	max heart rate
min_hr	min heart rate
std_hr	standard deviation of heart rate

**Table 5 sensors-22-06554-t005:** Quantitative comparison of the proposed method (2D-CNN + Channel-wise Attention) and the baseline methods (traditional machine learning), using the original 60 s window length ECG as input. The results are presented as mean ± standard deviation. The best performance is indicated in bold.

	**60 s Window Length Log-Spectrogram**					
**Method**	**F1 Score**	**Precision**	**Recall**	**Specificity**	**Accuracy**	**AUC**
2D-CNN + Channel-wise Attention	0.79 ± 0.03	**0.78 ± 0.08**	0.82 ± 0.05	0.85 ± 0.08	0.84 ± 0.04	**0.84 ± 0.03**
	**No Time Series Images**					
**Method**	**F1 Score**	**Precision**	**Recall**	**Specificity**	**Accuracy**	**AUC**
Random Forest (HRV time domain features)	**0.81 ± 0.00**	0.77 ± 0.00	**0.85 ± 0.01**	**0.85 ± 0.00**	**0.85 ± 0.00**	0.80 ± 0.00

## Data Availability

Data not publicly available due to ethical restrictions.

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
