# Peer review of "Classification of Tetanus Severity in Intensive-Care Settings for Low-Income Countries Using Wearable Sensing"

_sensors, 2022, doi:10.3390/s22176554_

Round 1

Reviewer 1 Report

The authors have applied neural network techniques for early detection of tetanus from low-cost wearable sensors data. They have explored different techniques with deep learning and shown that a two-dimensional (2D) convolutional neural network with a channel-wise attention mechanism works best. They have worked on patient data from day 1 and day 5. But it would make more sense if the authors would only work on day 1 data, not on day 5 data as they are aiming for early detection of tetanus. 

 Please explain the following questions and revise the manuscript accordingly:

1. Why is specifically the log spectrogram used for converting the 1D ECG data?

2. Did the authors cross validate the data while splitting into training, validation and test phase for evaluating the performance metric (accuracy, F1 score etc.)?

3. The table number is missing in line 336. 

4. It can be seen from the results that, 50 seconds time window has a comparatively higher value of performance metrics than other window length. Is there any explanation for this? 

Author Response

Point 1: Why is specifically the log spectrogram used for converting the 1D ECG data?

Response 1: The visible image patterns in Spectrograms are hard to see.    With a log scale, the low frequencies are easier to see using Log Spectrograms. Hence, the image patterns in Log Spectrograms  image patterns enhances understanding of 2D CNNs.

Point 2:  Did the authors cross validate the data while splitting into training, validation and test phase for evaluating the performance metric (accuracy, F1 score etc.)?

Response 2: Thank you for asking this question. In our experiments, we did not use cross validation.  Instead, we ran each model five times with the same split training/ validation/ test dataset,  because our dataset is large enough to split into training/ validation/ test dataset. There are 178 ECG examples from 110 patients on days 1 and 5. Then we chose 30 one-minute ECG samples from each ECG example file. There are 4230 one-minute ECG samples in the training set, 540 one-minute ECG samples in the validation set, and 570 one-minute ECG samples in the test set. Also, the number of severe disease and mild disease are quite balanced. Moreover, ECG samples are independent.

Point 3: The table number is missing in line 336. 

Response 3: Thank you for pointing this out. The table number has been added.

Point 4:  It can be seen from the results that, 50 seconds time window has a comparatively higher value of performance metrics than other window length. Is there any explanation for this? 

Response 4: Thank you for mentioning this. We investigate the window length of the Spectrogram and investigate the range of window lengths that can maintain an accurate tetanus diagnosis. In our experiments, a 50s window has a relatively higher value of performance metrics than other window lengths. We will explore time series imaging further in future work, which will aim to find the optimal range of time windows

Reviewer 2 Report

The manuscript by Ping Lu et al., is an interesting study looking at classification of Tetanus severity based on long term ECG monitoring using a deep learning approach. Although the concept is interesting and potential useful for triage in low income places there are several major comments about the study which the authors need to address.

- if I understood the study well, the authors are performing classification not prediction. I invite the authors to understand the difference and use only the one more suited to the method as they are using both a long the manuscript and the title can be misleading. I figure or diagram of the methodology will also help the reader.

- As part of the motivation, the authors suggest ANS which is commonly measure by HRV as part of the mechanisms associated to the use of ECG for Tetanus classification. They mentioned that computing HRV is expensive, this is not correct as there are free validated toolboxes that compute HRV based on ECG raw signals, such as: https://www.physionet.org/content/pcst/1.0.0/,  https://marcusvollmer.github.io/HRV/, and https://hrvtoolkit.com/, and their computational time is not higher as running a deep learning method. I invite the authors to perform a comparison study which uses standard HRV metrics for the analysis of ANS.

- Similar to the previous point, from the figures/spectrograms, it seems like the first layers of the neural network are just filtering the data and extracting RR and respiration information which is later used in the linear layers (Figure 1). A comparison vs simple machine learning models with HRV metrics, including some feature engineering would provide a better insight on the ANS dysfunction mechanism.

- It seems like the authors just want to create a deep learning approach as they do not want to understand the mechanisms behind HRV and ANS, including the relation of these two in the introduction will help the authors and reader to better understand the motivation behind using ECG signals.

- For the machine learning approach, the number of subjects is very low, I would strongly suggest the authors to use K-fold cross validation and present the average results for all the different models given the low number of subjects and potential overfitting of their model due to it.

- I looks like most references are either websites, conference proceedings, or arXiv preprints. I would invite the authors to do a more extensive literature review to include per-reviewed full research studies.

Author Response

Point 1:  if I understood the study well, the authors are performing classification not prediction. I invite the authors to understand the difference and use only the one more suited to the method as they are using both a long the manuscript and the title can be misleading. I figure or diagram of the methodology will also help the reader.

Response 1: Thank you for your comments. I think the title is not clear enough. The word classification would be better to explain this work, instead of the word predicting. The updated title is Classification of Tetanus Severity in Intensive-Care Settings for Low-Income Countries using Wearable Sensing. In this work, we use day 1 and day 5 for classification. The future work will only use day 1 data to predict severe /mild tetanus disease on day 5. An illustration of the channel-wise attention mechanism structure has been added to describe the mechanism.

Point 2:  As part of the motivation, the authors suggest ANS which is commonly measure by HRV as part of the mechanisms associated to the use of ECG for Tetanus classification. They mentioned that computing HRV is expensive, this is not correct as there are free validated toolboxes that compute HRV based on ECG raw signals, such as:

https://www.physionet.org/content/pcst/1.0.0/,  https://marcusvollmer.github.io/HRV/, and https://hrvtoolkit.com/, and their computational time is not higher as running a deep learning method. I invite the authors to perform a comparison study which uses standard HRV metrics for the analysis of ANS

Response 2: Thank you for your advice. I have deleted the sentence in the manuscript “However, its detection requires expensive equipment and expertise which is usually not available.”. Also, I added one section about the traditional machine learning. In our work, we detect r peaks of ECG using the open-source packages 354 py-ecg-detectors 1.3.2 [48] and extract features using the open-source packages hrv-analysis 355 1.0.4 [47]. We explore HRV time domain features and HRV frequency domain features.

Point 3: Similar to the previous point, from the figures/spectrograms, it seems like the first layers of the neural network are just filtering the data and extracting RR and respiration information which is later used in the linear layers (Figure 1). A comparison vs simple machine learning models with HRV metrics, including some feature engineering would provide a better insight on the ANS dysfunction mechanism.

Response 3: We added experiments for traditional machine learning. We chose random forest for Classification of Tetanus Severity.  The random forest models are used HRV time domain features, HRV frequency domain features, HRV time & frequency domain features respectively.

Point 4:  It seems like the authors just want to create a deep learning approach as they do not want to understand the mechanisms behind HRV and ANS, including the relation of these two in the introduction will help the authors and reader to better understand the motivation behind using ECG signals.

Response 4: Thank you for your comments. I added the traditional machine learning method - random forest – as baseline.

Point 5:  For the machine learning approach, the number of subjects is very low, I would strongly suggest the authors to use K-fold cross validation and present the average results for all the different models given the low number of subjects and potential overfitting of their model due to it.

Response 5: Thank you for your advice. In our experiments, we do not use cross validation.  In our experiments, we run each model five times with the same split training / validation / test datasets, because our spectrogram dataset is large enough to split into these datasets. Also, the number of severe disease and mild disease cases are quite balanced. Moreover, one-minute ECG samples are independent. There are 178 ECG examples from 110 patients on day 1 and 5. Then we choose 30 one-minute ECG samples from each ECG example file. There are 4230 one-minute ECG samples in the training set, 540 one-minute ECG samples in the validation set, and 570 one-minute ECG samples in the test set. Also, the number of severe disease and mild disease cases are quite balanced. Moreover, one-minute ECG samples are independent.

Point 6:  I looks like most references are either websites, conference proceedings, or arXiv preprints. I would invite the authors to do a more extensive literature review to include per-reviewed full research studies.

Response 6: Thank you for your advice. I added several literature references in my work. In the computer vision field, the latest technologies are usually published in conference proceedings. For example, Squeeze-and-excitation networks from Proceedings of the IEEE conference on computer vision and pattern recognition (citation number14435). Also, some works in arXiv preprints are the state-of-the-art in my research field. For example, the arXiv paper “An image is worth 16x16 words: Transformers for image recognition at scale” are from Google Research, Brain Team. The citation number of this paper is 6017. People working on transformer in imaging normally cite this paper in their own works.

Round 2

Reviewer 2 Report

The authors had answer most of my comments, however, there is still one that they are failing to address and I believe it is an important issue.

- The authors mentioned the do not perform cross validation because the spectrogram dataset is large enough. However, they only have 178 ECG examples from 110 patient. And they mention the one-minute ECG samples are independent but they are not as they are taken from the same patient, at the same position just a few minutes difference. Therefore, the ECG will have most likely the same morphology with small variations in HR. More importantly, has any ECG from a subject used in training was also part of the validation and test. As the one-minute ECG samples are not independent authors cannot use the data from the same subject from the training in the validation or test set. Unless, the authors are looking for patient specific algorithms to predict recovery, which in case they also have to modify their methods design.

- Interestingly, the F1 score was higher for the random forest with HRV metrics, please add a paragraph in the discussion about it.

Author Response

Point 1: The authors mentioned the do not perform cross validation because the spectrogram dataset is large enough. However, they only have 178 ECG examples from 110 patient. And they mention the one-minute ECG samples are independent but they are not as they are taken from the same patient, at the same position just a few minutes difference. Therefore, the ECG will have most likely the same morphology with small variations in HR. More importantly, has any ECG from a subject used in training was also part of the validation and test. As the one-minute ECG samples are not independent authors cannot use the data from the same subject from the training in the validation or test set. Unless, the authors are looking for patient specific algorithms to predict recovery, which in case they also have to modify their methods design.

Response 1: The reviewer is right. The 60s ECG samples are not independent. We have now removed this from our manuscript. However, we perform the splitting of the dataset into training, validation and test based on unique ECG samples. After splitting, we apply windowing on ECG time series to split each signal into 60 second time series. Therefore, we made sure that ECG samples in each split of the dataset are unique.

Point 2:  Interestingly, the F1 score was higher for the random forest with HRV metrics, please add a paragraph in the discussion about it.

Response 2: Thank you very much for pointing it out. I checked my code for Random Forest. I found that I misused a small dataset which is used to debug my code. Hence, I run the code of Random Forest again with the correct dataset which is used for deep learning. However, I found the open-source package is not able to extract frequency domain features in one 60s ECG sample correctly. Hence, I remove the Random Forest with frequency domain features and the Random Forest with time & frequency domain features. In my future work, I will explore different time & frequency domains, and try out the most important features.